# Use of community engagement interventions to improve child immunisation in low-income and middle-income countries: a systematic review and meta-analysis

Monica Jain ,[1] Shannon Shisler,[2] Charlotte Lane ,[3] Avantika Bagai,[4] Elizabeth Brown,[5] Mark Engelbert  [3]

[1]International Initiative for Impact Evaluation, New Delhi, India
[2]International Initiative for Impact Evaluation, London, UK
[3]International Initiative for Impact Evaluation, Washington, District of Columbia, USA
[4]Development Solutions, New Delhi, India
[5]Center for Effective Global Action, University of California, Berkeley, California, USA

**Correspondence to**
Dr Monica Jain;
mjain@3ieimpact.org

## ABSTRACT

**Objective** To support evidence informed decision-making, we systematically examine the effectiveness and cost-effectiveness of community engagement interventions on routine childhood immunisation outcomes in low-income and middle-income countries (LMICs) and identify contextual, design and implementation features associated with effectiveness.

**Design** Mixed-methods systematic review and meta-analysis.

**Data sources** 21 databases of academic and grey literature and 12 additional websites were searched in May 2019 and May 2020.

**Eligibility criteria for selecting studies** We included experimental and quasi-experimental impact evaluations of community engagement interventions considering outcomes related to routine child immunisation in LMICs. No language, publication type, or date restrictions were imposed.

**Data extraction and synthesis** Two independent researchers extracted summary data from published reports and appraised quantitative risk of bias using adapted Cochrane tools. Random effects meta-analysis was used to examine effects on the primary outcome, full immunisation coverage.

**Results** Our search identified over 43 000 studies and 61 were eligible for analysis. The average pooled effect of community engagement interventions on full immunisation coverage was standardised mean difference 0.14 (95% CI 0.06 to 0.23, $I^2$=94.46). The most common source of risk to the quality of evidence (risk of bias) was outcome reporting bias: most studies used caregiver-reported measures of vaccinations received by a child in the absence or incompleteness of immunisation cards. Reasons consistently cited for intervention success include appropriate intervention design, including building in community engagement features; addressing common contextual barriers of immunisation and leveraging facilitators; and accounting for existing implementation constraints. The median intervention cost per treated child per vaccine dose (excluding the cost of vaccines) to increase absolute immunisation coverage by one percent was US$3.68.

## STRENGTHS AND LIMITATIONS OF THIS STUDY

⇒ Thorough literature search of 21 major electronic databases and reporting as per Preferred Reporting Items for Systematic Reviews and Meta-Analyses guidelines.

⇒ Presents a nuanced framework of community engagement with a typology that differentiates three types of interventions: interventions in which community engagement is embedded, and those interventions that engage community in their design or implementation.

⇒ The effects of community engagement interventions are robust to exclusion of studies assessed as high risk of bias for almost all the primary outcomes. The effects are also uniform across geographies and baseline immunisation rates.

⇒ For some immunisation outcomes the evidence base for drawing conclusions is adequate, for others it is limited.

⇒ Evidence base is skewed across the three engagement types with a relatively large evidence base for those interventions in which community engagement is embedded and limited for those interventions using engagement in implementation autonomy

**Conclusion** Community engagement interventions are successful in improving outcomes related to routine child immunisation. The findings are robust to exclusion of studies assessed as high risk of bias.

## INTRODUCTION

Immunisation is one of the most cost-effective ways to prevent and control life-threatening infectious diseases. From 2001 to 2020, projects that introduced or increased coverage of vaccines averted an estimated 14 million deaths, 350 million cases of illness, 8 million cases of long-term disability and 700 million disability-adjusted life-years.[1] Nonetheless,

rates of routine vaccination of children in low-income and middle-income countries (LMICs) are low or stagnant. In 2019, an estimated 19.7 million infants did not receive routine immunisations. Around 60% of these children live in ten LMICs, including Ethiopia, India, Nigeria and Pakistan as of 2019.[2]

Community engagement approaches feature prominently in global immunisation strategies.[3] However, there is a dearth of rigorous and systematic evidence on effectiveness of community engagement approaches to improve routine childhood immunisations specifically in LMICs. In our search, we could find only two systematic reviews for LMICs, which analysed effectiveness of community monitoring interventions and preventive interventions delivered by community health workers, respectively.[4 5] As such, previous systematic reviews do not provide adequate guidance to stakeholders interested in understanding whether and how alternative community engagement interventions work in LMICs to improve routine childhood immunisations and at what cost. There is, therefore, a need to make such evidence available to guide policymakers and public health practitioners in making informed decisions about these interventions. To address this knowledge gap, we conducted a systematic review examining the effects of community engagement interventions on outcomes related to childhood immunisation in LMICs, determining their cost-effectiveness and identifying contextual, design and implementation features that may be associated with intervention effectiveness.

## METHODS
### Overview
The protocol of this systematic review with meta-analysis is registered with The Campbell Collaboration.[6] We followed the Campbell and Cochrane Collaborations' guidelines for systematic reviewing[7–10] and drew on theory-based mixed-methods impact evaluation[11] and systematic review[12 13] concepts. We followed the PRISMA reporting guidelines. The amendments to the information provided in the protocol is reported in online supplemental appendix 1.

### Conceptual framework
For our review, we defined 'communities' in reference to the lowest level of the health service delivery system (or whatever level provides routine immunisation services in the local context). A community is a group of people who serve or are served by a particular primary health facility. Thus, communities encompass a wide range of stakeholders, including caregivers, health service providers and influential community members such as religious or other traditional leaders.

WHO 2020 defines community engagement as 'a process of developing relationships that enable stakeholders to work together to address health-related issues and promote well-being to achieve positive health impact and outcomes.'[14]

For this review, we developed a framework that classified community engagement interventions based on process of engagement as in the WHO definition. It also corresponds to the 'utilitarian perspective' of community engagement captured and articulated in Brunton et al[15]: 'In utilitarian perspectives, health (and other) services reach out to engage particular communities that they have identified require assistance and the intervention is devised within existing policy, practice and resource frameworks.' In addition, our framework goes beyond one-way communication to include some consultation or dialogue with the community or some decision-making by them. We considered three points within an intervention during which engagement could occur, as elaborated below and in online supplemental appendix 2.

Engagement in the design of interventions: Community input or feedback was sought before implementing an intervention (eg, pilot, needs assessment, formative evaluation and outreach).

Engagement in implementation autonomy of interventions: Community was used in intervention implementation as healthcare workers, facilitators or problem solvers and only if they had some opportunity to affect or influence its implementation.

Engagement as the intervention (engagement is embedded): A serious attempt was made to gain community buy-in for activities or new cadres of community-based structures were established (eg, village health committees or community health volunteers).

### Research questions
The research questions for this review were:
1. What evidence exists regarding the effectiveness of community engagement interventions in improving routine immunisation coverage of children in LMICs?
2. Is there evidence for heterogeneous effects of community engagement strategies (ie, does effectiveness vary by geographical region, gender or programme implementation)?
3. What factors relating to programme design, implementation and context are associated with better or worse outcomes along the causal chain? Do these vary by the kind of community engagement?
4. What is the cost-effectiveness of different community engagement interventions in improving children routine immunisation outcomes?

### Search strategy
We implemented a systematic and comprehensive search strategy, in consultation with an information specialist. In May 2019 and May 2020, we searched 17 academic databases for experimental and quasi-experimental impact evaluations of community engagement interventions considering outcomes related to routine child immunisation in LMICs (using the World Bank country income classifications to determine LMIC status at the time the intervention began). We also searched 17 additional websites for grey literature. The list of sources searched and an example set of search strings are provided in

**Table 1** PICOS inclusion and exclusion criteria

| Characteristics | Inclusion criteria |
|---|---|
| Population | Rural, periurban and urban populations living in low-income and middle-income countries. |
| Interventions | Interventions involving community engagement. |
| Comparisons | A comparison group or counterfactual that does not receive the intervention or business as usual. |
| Outcomes | Full, partial, timely immunisation of children and other outcomes such as morbidity, mortality, etc. |
| Study design | To answer question 1, 2 and 4, experimental and quasi-experimental studies.<br>To answer question 3, qualitative studies, descriptive quantitative studies, process evaluations, project documents, formative research studies, protocols, baseline and midline and endline/final impact evaluation reports, policy briefs and website content. |
| Other | No inclusion restrictions by publication status or language. |

online supplemental appendix 3. We complemented this with citation tracking and contacting experts. The grey literature search was conducted by AB with support from external consultant reviewers. Given the limitations of the search functions on websites we searched for grey literature, it was not possible to use the same complex search strings used in academic databases, and search strategies were developed on a site-by-site basis.

### Inclusion/exclusion criteria (population, intervention, comparators, outcomes and study designs)

The population, intervention, comparators, outcomes and study designs eligible for inclusion in the study are provided in table 1. No language, publication type or date restrictions were imposed. Because our definition of community focused on the lowest levels of health facilities, we excluded interventions targeting higher levels of the health system (eg, state-level officials) (online supplemental appendix 4). The primary outcomes considered in this review were coverage rates for (A) full immunisation, which is typically defined as the percentage of 1 year old who have received one dose of Bacille Calmette-Guérin (In some countries, other vaccinations such as those for JE encephalitis and yellow fever are administered to children as a part of the routine immunisation schedule. In those contexts, we went by the definition of full immunisation mentioned in the impact evaluation study), (B) third dose of DPT or pentavalent, (C) first dose of measles or (D) the timeliness of any of these doses. Additional antigen-specific immunisation coverage outcomes and secondary outcomes reflecting upstream conditions (eg, attitudes about vaccination and access to immunisation services) and downstream effects (eg, morbidity and mortality) of the primary outcomes were also included. Official health records and parent recall were considered acceptable sources of measures of immunisation coverage. The former was used when both measures were reported separately.

This review includes experimental and quasi-experimental studies that estimate the causal impact of an intervention, as compared with usual practice, by establishing a counterfactual. Specifically, studies with the following evaluation designs are included: randomised controlled trials, regression discontinuity designs,

instrumental variables' estimation, statistical matching (eg, propensity score matching), difference-in-differences (or any mathematical equivalent), fixed effects estimation and interrupted time series. We excluded studies for which the reported quantitative data could not be meaningfully converted to an effect size. In cases of relevant missing or incomplete data, we contacted study authors to obtain the required information. If we were unable to obtain the necessary data, we reported the characteristics of the study but did not include these studies in the meta-analysis. We conducted additional searches for economic and qualitative evidence on the included impact evaluations (online supplemental appendix 5).

### Screening

At both the title and abstract and full-text screening stages, all papers were double screened by research consultants and supervised by MJ, ME and AB. Reconciliation meetings were held to resolve disagreements, and MJ and ME made final decisions on unresolved cases. The same reviewers manually searched for qualitative papers and project documents on Google Scholar and websites of implementing organisations and screened the papers for inclusion as they were identified.

### Data analysis

Studies were coded for their engagement type by two reviewers (MJ and AB) who independently reviewed the intervention description and coded these against the definitions provided above. If studies allowed for engagement at several stages of the intervention, they could be coded as having more than one engagement type. We used Microsoft Excel to extract descriptive information and effect sizes from included studies using double coding. Coders reconciled their answers, and a study author made final decisions in case of disagreements. For qualitative analysis, all impact evaluations and additional documentation identified in the search were coded in NVivo. Cost data were single coded and checked by a study author (online supplemental appendix 6).

To avoid double-counting of evidence from different papers focusing on the same study, we linked these papers prior to analysis. We extracted data from the most recent publication. When data were reported over multiple time

periods, we extracted data for each period. Where authors reported the same outcome using more than one analytical model, we extracted data from the authors' preferred model specification. When the preference was not specified, we used the model with the most controls. Where studies reported outcomes related to multiple treatment arms and only one comparison group, we estimated an effect size for each of the treatment arms.

To assess quantitative risk of bias, we created an adapted version of the Cochrane guidelines for assessing randomised controlled trials and non-randomised studies.[16 17] These assessments were conducted by two independent reviewers. Coders reconciled their answers, and a study author made final decisions in case of disagreements. For testing the sensitivity of the results to low-quality studies, we ran each analysis with and without studies scoring high risk of bias.

We critically appraised qualitative and mixed-methods studies using an adapted version of the nine-item framework developed by the Critical Appraisal Skills Programme.[18] In addition, we carried out a sensitivity analysis in which we considered only the high-quality qualitative studies that had a risk of bias assessment score of 20 or higher, indicating low risk of bias. For cost evidence, we assessed risk of bias along six primary dimensions adapted from a combination of tools, including: Doocy and Tappis[19]; Campbell Collaboration Economic Methods Policy Brief[20] and Methods for the Economic Evaluation of Health Care Programmes[21] (online supplemental appendix 7).

We calculated the standardised mean difference, or Cohen's d, its variance and SE for each effect, converting effects reported in other metrics as necessary, using formulae provided in Borenstein *et al.*[22] In all cases we then adjusted Cohen's d to Hedges' *g* as defined in Ellis.[23] For studies reporting regression results, we followed the approach of Keef and Roberts[24] using the regression coefficient and the pooled SD of the outcome.

The amount of heterogeneity (ie, $\tau^2$) was estimated using the DerSimonian-Laird estimator.[25] The *Q*-test for heterogeneity[26] and the $I^2$ statistic[27] are reported. We complement this with an assessment of heterogeneity of effect sizes graphically using forest plots. We identified outliers using studentized residuals and identified overly influential studies using Cook's distance. Where outliers were indicated, we report the resulting effect sizes when they are left out of the analysis. As an additional sensitivity test, we ran a full leave-one-out analysis for all models, and we report these results when and where they are useful. Whenever feasible, we conducted moderator analyses using meta-regression to investigate sources of heterogeneity. (All but two moderators were chosen a priori. Baseline coverage and vaccine hesitancy were added after feedback from an initial peer review from the Campbell Collaboration (copublisher of this work).) The analysis was carried out using R (V.4.0.4)[28] and the metafor package (V.2.4.0).[29] All analyses used a random effects model because we did not reasonably expect the

included studies to be functionally identical and the goal was to generalise to the larger population.[30]

Qualitative analysis followed a mix of inductive and deductive coding approaches to identify themes related to barriers and facilitators, reasons for intervention success or failure, and uptake and fidelity challenges. An initial set of themes was developed based on familiarity with the literature. However, as new topics were identified, new themes were added. Themes were also disaggregated if it became clear they were too broad. Research consultants conducted coding with oversight from CL and AB.

### Patient and public involvement in research

There was no patient or public involvement in this research.

## RESULTS

Our search identified over 43 000 records, which were reduced to 29 481 unique abstracts after deduplication (figure 1). After title and abstract screening, we considered 1285 studies for full-text screening and could not locate an additional 44, published mostly before 2000. We excluded articles at full-text for not satisfying the inclusion criteria by country (129), study type (evaluation study) (304), evaluation method (213), outcome (235) and community engagement type (172). We ultimately identified 61 impact evaluations (table 2)[31–92] that assessed the effects of community engagement interventions on outcomes related to routine child immunisation in LMICs. We identified one publication in Spanish,[45] with all others in English. Five studies did not include sufficient data to calculate an effect size, thus, 56 studies were included in the meta-analysis. Inter-rater reliabilities were calculated on a sample of studies, and ranged from 28% (mean effect of the intervention to 100% (eg, country, publication year and study design). All studies were reconciled prior to analyses.

### Risk of bias

Of the 31 included studies with experimental designs, only two had a low risk of bias, six had some concerns and 23 had a high risk of bias. Of the 30 included quasi-experimental studies, only 2 were assessed as low risk of bias, 1 as some concerns and 27 as high risk of bias. Biases arising from outcome measurement and deviations from intended interventions were the most common across both the study designs.

Although only five qualitative studies scored strong on all key elements, most studies received strong scores on most key elements and had quality appraisal scores greater than 20, indicating low risk of bias. The most common elements found to be missing were sample characteristics and analytical methods. The quality of the cost evidence in the 22 studies that included such evidence was mixed (further information and visualisations for the risk of bias appraisals can be found in online supplemental appendix 8).

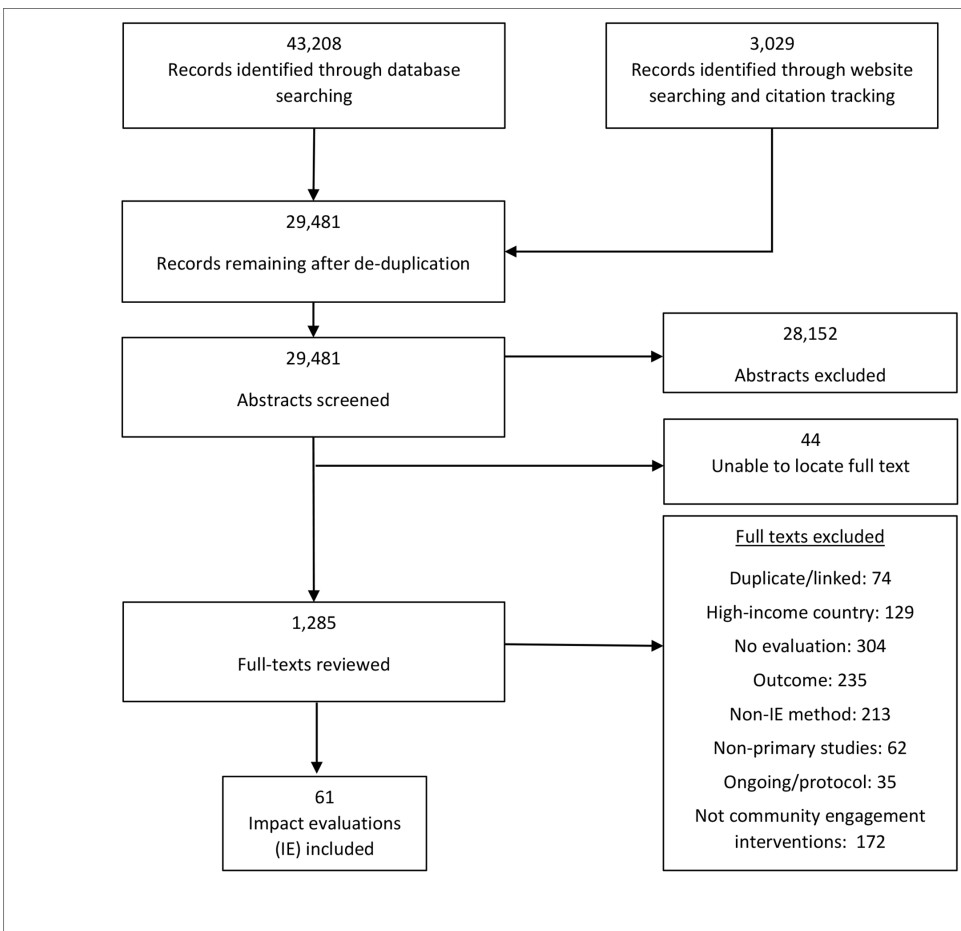

**Figure 1** PRISMA flow diagram. PRISMA, .Preferred Reporting Items for Systematic Reviews and Meta-Analyses.

## Community engagement interventions

### Full immunisation

A total of $k = 28$ studies examined the effect of community engagement interventions on full childhood immunisation and found $\hat{\mu} = 0.14$ (95% CI 0.06 to 0.23), z = 3.28, p = 0.01), indicating a small but significant benefit for the treated group of 0.14 SD units (figure 2). A 95% credibility/prediction interval for the true outcomes is given by −0.28 to 0.57. Hence, although the average outcome is estimated to be positive, in some studies the true outcome may in fact be negative.

The rank correlation test indicated funnel plot asymmetry ($p = 0.03$) but the regression test did not ($p = 0.57$ ; see online supplemental appendix 9 for additional information). The true effects appear to be heterogeneous ($I^2 = 94.5\%$, $\tau = 0.21$). Outlier analyses revealed that Banerjee may be a potential outlier, and sensitivity analyses removing Banerjee (2010) reduced the overall average effect ($\mu = 0.08$ (95% CI 0.04 to 0.12)), but it was still positive and significant ($z = 4.12$, $p < 0.001$). (For all other outcomes, outlier analyses will be presented in online supplemental appendices.) Sensitivity analysis using the leave-one-out approach indicates there are no other studies whose removal results in substantial changes to the average effect or overall heterogeneity.

Sensitivity analyses were also conducted to examine the robustness of the results to the exclusion of low-quality studies. When studies assessed as high risk of bias were removed, the resulting effect was slightly larger and still statistically significant ($\hat{\mu} = 0.18$ (95% CI 0.08 to 0.27)), $k = 4$, $z = 3.67$, $p < 0.001$. We examined several potential sources of heterogeneity, including exposure to the intervention, evaluation period, study design, year, geographical region, data source, whether the intervention was implemented by a government agency (either alone or in combination with another agency), whether new cadres of health workers were established, presence of vaccine hesitancy and baseline vaccine coverage rates. There were no significant moderators in the context of this model (see online supplemental appendix 9).

### DPT 3

A total of $k = 22$ studies examined the effect on DPT3 vaccination coverage and found a small but significant benefit to the treated group compared with the untreated group ($\hat{\mu} = 0.10$ (95% CI 0.06 to 0.14, z = 4.75), $p < 0.001$ ; figure 3). A 95% credibility/prediction interval for the true outcomes is given by −0.06 to 0.26. Hence, although the average outcome is estimated to be positive, in some studies the true outcome may in fact be negative.

**Table 2** Describing characteristics of the included studies

| Author | Country | Summary of intervention | Engagement type | Study design | Duration (in months) | Outcomes |
|---|---|---|---|---|---|---|
| Admassie et al 2009[31] | Ethiopia | Formation of a cadre of community-based health extension workers and using community resources for construction of health posts.<br>The study evaluates intervention effectiveness and cost-effectiveness. | Multiple (EII+EAI) | PSM | 48 | FIC, BCG, DPT3, OPV1, OPV2, OPV3, measles, morbidity |
| Adamu et al 2019[32] | Nigeria | Quality improvement programme where health workers use iterative processes to develop localised and contextually relevant plans to resolve health service delivery and demand bottlenecks.<br>The study evaluates intervention effectiveness. | Multiple (EID+EII) | ITS | 1 | Dropouts |
| Alhassan et al 2019[33] | Ghana | Using a bottom-up approach, the intervention recruited and trained community groups to identify service delivery gaps in healthcare facilities.<br>The study evaluates intervention effectiveness. | Multiple (EII+EAI) | RCT | 10 | FIC |
| Andersson et al 2009[34] | Pakistan | Community dialogues to address barriers to vaccination. The guidelines for the dialogue were created after consultation with the intended beneficiaries.<br>The study evaluates intervention effectiveness and cost-effectiveness. | Multiple (EID+EAI) | RCT | 8 | Knowledge about immunisation, attitude about immunisation, community norms, readiness to vaccinate, household norms and decision-making measles, DPT3 |
| Arifeen et al 2009[35] | Bangladesh | Formation of a cadre of village health volunteers and enlisting support of local religious leaders to convey messages about child health.<br>The study evaluates intervention effectiveness. | EAI | RCT | 71 | Measles, mortality |
| Assegaai et al 2018[36] | South Africa | Lay community-based workers were formalised as community health workers and served as a part of the outreach teams.<br>The study evaluates intervention effectiveness. | EAI | DID | 36 | FIC, measles, morbidity |
| Banerjee et al 2010[37] | India | Provision of immunisation services and incentives to caregivers. A trusted community-based organisation was a key stakeholder in design and delivery of the intervention.<br>The study evaluates intervention effectiveness and cost-effectiveness. | Treatment 1: EAI<br>Treatment 2: Multiple (EID+EAI) | RCT | 18 | FIC, BCG, partial immunisation |
| Banerjee et al 2020[38] | India | This evaluation tested two different interventions:<br>1. Incentives to caregivers. The community's feedback was solicited on the kind of incentive.<br>2. Community influencers were identified to spread information about immunisation.<br>The study evaluates intervention effectiveness and cost-effectiveness. | EID<br>EAI | RCT | 14 | Knowledge about immunisation, attitude about immunisation, FIC, DPT1, DPT2, DPT3, measles |
| Banwat et al 2015[39] | Nigeria | Female members of the community whose children are fully immunised were nominated in each community to serve as peer educators.<br>The study evaluates intervention effectiveness. | EAI | CBA | – | FIC, knowledge about immunisation, attitude about immunisation, readiness to vaccinate |
| Biemba et al 2016[40] | Zambia | A national policy to create a cadre of well-trained and motivated community-based health workers.<br>The study evaluates intervention effectiveness. | EAI | DID | 23 | FIC, morbidity |
| Björkman et al 2009[41] | Uganda | Communities were involved in monitoring the quality of health services and the performance of health service providers.<br>The study evaluates intervention effectiveness and cost-effectiveness. | EII | RCT | 0.16 | FIC, BCG, OPV0, OPV3, DPT1, DPT3, measles, Partial routine immunisation, mortality |
| Bolam et al 1998[42] | Nepal | Training for community health workers and midwives which was developed in collaboration with health workers and experts.<br>The study evaluates intervention effectiveness. | EID | RCT | 3 | FIC |
| Borkum et al 2014 Carmichael et al 2019 (linked study)[43 44] | India | Performance-based incentives to frontline workers. The nature of incentives was decided on in consultation with the frontline workers.<br>The study evaluates intervention effectiveness and cost-effectiveness. | EID | RCT | 12 | Health card availability, CHW capacity, FIC, BCG, DPT1, DPT2, DPT3, OPV1, OPV2, OPV3, measles, partial immunisation, timeliness |
| Calderón-Ortiz and Mejía-Mejía 1996[45] | Mexico | Creation of a community-based cadre of volunteers to register and track children in the community for immunisation.<br>The study evaluates intervention effectiveness. | EAI | CBA | 4 | FIC, BCG, DPT3, OPV3, measles |
| Carnell et al 2014[46] | Ethiopia | Formation of a cadre of community health workers to mobilise the community and encourage uptake of health services.<br>The study evaluates intervention effectiveness and cost-effectiveness. | EAI | DID | 60 | DPT3, measles |

Continued

**Table 2** Continued

| Author | Country | Summary of intervention | Engagement type | Study design | Duration (in months) | Outcomes |
|---|---|---|---|---|---|---|
| Costa-Font et al 2017[47] | India | Establishment of the village health and sanitation committees to monitor health service provision at the community level.<br>The study evaluates intervention effectiveness. | EAI | IV | – | BCG, DPT1, OPV0 |
| Demilew et al 2020[48] | Ethiopia | A poster/stamp system that reminded health workers of the child's immunisation status and simultaneously encouraged caregivers to immunise their children. The intervention was designed in consultation with health workers.<br>The study evaluates intervention effectiveness and cost-effectiveness. | Multiple (EID+EAI) | RCT | 17 | FIC, BCG, DPT1, DPT2, DPT3, partial immunisation |
| Dipeolu 2017[49] | Nigeria | Text message reminders to mothers regarding immunisation schedule. The messages were field tested with mothers to get the content right.<br>The study evaluates intervention effectiveness. | EID | DID | 9 | Knowledge about immunisation, attitude about immunisation, timeliness |
| Domek et al 2019[50] | Guatemala | SMS text messages to caregivers. A prior feasibility and acceptability study was conducted for the intervention.<br>The study evaluates intervention effectiveness. | EID | RCT | 2 | Timeliness |
| Engineer et al 2016[51] | Afghanistan | Pay-for-performance bonuses paid quarterly to health workers. The bonus amount was revised after receiving health worker feedback.<br>The study evaluates intervention effectiveness. | EID | RCT | 24 | Experience & satisfaction with health services, Formal HW motivation, capacity & performance, DPT3 |
| Findley et al 2013[52] | Nigeria | Formation of a cadre of community volunteers to facilitate group discussions on health and track/register women/children for health services.<br>The study evaluates intervention effectiveness and cost-effectiveness. | EAI | DID | 24 | FIC |
| Gibson et al 2017[53] | Kenya | SMS reminders and monetary incentives to caregivers. A feasibility study was conducted in 2013 for this intervention.<br>The study evaluates intervention effectiveness. | EID | RCT | 12 | Community norms, FIC, BCG, DPT1, DPT2, DPT3, OPV0, OPV1, OPV2, measles, timeliness |
| Goel et al 2012[54] | India | A multi-component campaign which involved women groups in awareness generation to improve health service uptake.<br>The study evaluates intervention effectiveness. | EAI | DID | 48 | FIC |
| Gurley et al 2020[55] | India | Community members were trained to design and produce culturally appropriate, 'hyperlocal' videos to promote health seeking behaviours.<br>The study evaluates intervention effectiveness and cost-effectiveness. | Multiple (EID+EAI) | RCT | 11 | Knowledge about immunisation, attitude about immunisation, FIC, DPT3, partial immunisation, timeliness, dropouts |
| Herrera-Almanza and Rosales-Rueda 2018[56] | Madagascar | Community-based primary healthcare services intervention that included the deployment of volunteer community health workers in remote areas.<br>The study evaluates intervention effectiveness. | Multiple (EII+EAI) | DID | 26 | Health card availability, OPV3, DPT3, measles, mortality, partial immunisation |
| Igarashi et al 2010[57] | Zambia | The GMP+sessions were conducted by medical personnel from Public Health Centres. During these sessions, community volunteers provided some operational and managerial support to ensure the effective implementation of the sessions.<br>The study evaluates intervention effectiveness. | EII | CBA | 43 | FIC, timeliness, attitude about immunisation, community norms |
| Janssens 2011[58] | India | Dissemination of health promoting messages to women in the community who are encouraged to further spread the awareness.<br>The study evaluates intervention effectiveness. | EAI | IV | 56.4 | DPT3, measles |
| Johri et al 2020[59] | India | Interventions, designed through formative research, to increase caregiver knowledge and adherence to childhood immunisation.<br>The study evaluates intervention effectiveness. | EID | RCT | 3 | Knowledge about immunisation, Awareness of place, time, schedule for vacc., attitude about immunisation |
| Lee 2015[60] | Zambia | Creation of a new cadre of frontline workers from the community, called community health assistants, to provide primary healthcare services.<br>The study evaluates intervention effectiveness. | EAU | RCT | 3 | Formal HW motivation, capacity & performance, BCG, OPV3, measles, timeliness, CHW capacity, morbidity |
| Mayumana et al 2017[61] | Tanzania | Payment-for-performance scheme for health facilities. Health workers and health facility governing committees decided the allocation of funds.<br>The study evaluates intervention effectiveness. | EII | DID | 30 | Stockouts |

**Table 2** Continued

| Author | Country | Summary of intervention | Engagement type | Study design | Duration (in months) | Outcomes |
|---|---|---|---|---|---|---|
| Memon *et al* 2015[62] | Pakistan | Formation of community health committees to promote perinatal and new-born care. Formative research informed the intervention design.<br>The study evaluates intervention effectiveness. | Multiple (EID+EAI) | DID | 16 | FIC |
| Modi *et al* 2019[63] | India | The mHealth intervention package consisting of mobile phone-based job aids for community health workers. The intervention was piloted in 2015.<br>The study evaluates intervention effectiveness and cost-effectiveness. | EID | RCT | 12 | DPT3, morbidity |
| Mohanan *et al* 2020[64] | India | Social accountability interventions to promote community-based collective action to improve delivery of health and nutrition services to children.<br>The study evaluates intervention effectiveness and cost-effectiveness. | EAI | RCT | 12 | Experience & satisfaction with health services, attitudes about health providers, formal health worker supply, FIC, BCG, DPT3, OPV1, OPV3, IPV, measles, morbidity, mortality |
| More *et al* 2012[65] | India | Urban slum-dweller women's groups used community dialogues to address barriers to improving perinatal health.<br>The study evaluates intervention effectiveness. | EAI | RCT | 36 | Mortality |
| More *et al* 2017[66] | India | The intervention comprised multiple activities like home visits to caregivers, groups meetings, community events and other supportive services.<br>The study evaluates intervention effectiveness and cost-effectiveness. | EAI | RCT | 24 | Health card availability, attitudes about health providers, FIC, BCG, measles, partial immunisation |
| Morris *et al* 2004[67] | Honduras | Monetary vouchers to women in the communities and setting up of community-based committees to oversee health service quality and access.<br>The study evaluates intervention effectiveness and cost-effectiveness. | EAI | RCT | 24 | DPT1, measles |
| Murthy *et al* 2019[68] | India | Voice call reminders to pregnant women and caregivers. The message content was tested for appropriateness through community focused groups.<br>The study evaluates intervention effectiveness. | EID | RCT | 21 | Knowledge about immunisation, FIC |
| Nagar *et al* 2018[69] | India | A digital pendant-based health record of the child and a voice call reminder system. A formative study was conducted in 2016 and communities were consulted on the design of the pendant.<br>The study evaluates intervention effectiveness. | EID | RCT | 3 | Timeliness |
| Nagar *et al* 2020[70] | India | A digital pendant-based health record of the child. Health providers used a mobile application to scan the pendant to update the child's medical history. Prior formative research informed the intervention design.<br>The study evaluates intervention effectiveness and cost-effectiveness. | EID | RCT | 20 | FIC, DPT1, DPT2, DPT3 |
| Nzioki *et al* 2017[71] | Kenya | Formation of a cadre of community health workers.<br>The study evaluates intervention effectiveness. | EAI | CBA | 0 | FIC |
| Oche *et al* 2011[72] | Nigeria | Group meetings with caregivers and dialogues with community leaders to improve uptake of routine immunisation services.<br>The study evaluates intervention effectiveness. | EAI | DID | 9 | Knowledge about immunisation, DPT1, DPT3, dropouts |
| Okeke *et al* 2017[73] | Nigeria | A national scheme to create, train and deploy a cadre of midwives to serve underserved rural and remote populations in Nigeria.<br>The study evaluates intervention effectiveness. | Multiple (EII+EAI) | DID | 40 | BCG, DPT3, OPV3, measles, mortality |
| Okoli *et al* 2014[74] | Nigeria | A conditional cash transfer programme to encourage uptake of health services. Community groups were consulted while deciding the cash amount.<br>The study evaluates intervention effectiveness. | EID | ITS | seven to 18 | OPV0 |
| Olayo *et al* 2014[75] | Kenya | Formation of a cadre of community health workers who then facilitated dialogue at the community level and supported other community-based workers.<br>The study evaluates intervention effectiveness. | Multiple (EII+EAI) | DID | 24 | Health card availability, DPT1, DPT3, measles |
| Olken *et al* 2014[76] | Indonesia | Block grants for maternal and child health that incorporated relative performance incentives were implemented in villages through creation of village-level health committees.<br>The study evaluates intervention effectiveness and cost-effectiveness. | Multiple (EII+EAI) | RCT | 18 to 30 | FIC, morbidity, mortality |

Continued

**Table 2** Continued

| Author | Country | Summary of intervention | Engagement type | Study design | Duration (in months) | Outcomes |
|---|---|---|---|---|---|---|
| Oyo-Ita et al 2020[77] | Nigeria | A multi-component intervention involving traditional and religious leaders for engaging communities in planning and delivery of immunisation services.<br>The study evaluates intervention effectiveness and cost-effectiveness. | Multiple (EII+EAI) | RCT | 18 | FIC, partial immunisation, Timeliness |
| Pramanik et al 2020[78] | India | Trained facilitators from local NGOs interacted with the communities to enable them to leverage their own strengths for addressing their concerns related to child health.<br>The study evaluates intervention effectiveness and cost-effectiveness. | EAI | RCT | 13 | Knowledge about immunisation, attitude about immunisation, attitudes about health providers, health card availability, FIC, DPT1, DPT2, DPT3, timeliness, dropouts |
| Rahman et al 2008[79] | Pakistan | Mental health support programme with counselling sessions for pregnant and post-partum women. Prior intervention pilots informed the intervention design.<br>The study evaluates intervention effectiveness. | EID | RCT | 11 | FIC, morbidity |
| Rahman et al 2016[80] | Bangladesh | New cadre of community health workers delivered essential maternal, neonatal and child healthcare and nutrition services.<br>The study evaluates intervention effectiveness. | EAI | DID | 48 | FIC, morbidity |
| Rao 2014[81] | India | Creation of a cadre of community health workers to improve basic health outcomes through community engagement.<br>The study evaluates intervention effectiveness. | EAI | DID | 60 | BCG, DPT3, OPV3, measles, FIC, partial immunisation, supply of CHWs |
| Robertson et al 2013[82] | Zimbabwe | Use of cash transfers for behaviour change. Local NGO and community leaders were involved in beneficiary targeting and compliance monitoring. The intervention was also tested for feasibility during a prior study.<br>The study evaluates intervention effectiveness. | Multiple (EID+EII) | RCT | 12 | FIC, community norms, OPV0 |
| Roy et al 2008[83] | Bangladesh | Rural maintenance programme recruited and trained women for road maintenance, health awareness, numeracy, human rights, gender equity, health and nutrition, and business management. | EAI | DID | 11 | FIC, BCG, DPT1, DPT2, DPT3, OPV0, OPV1, OPV2, OPV3, measles, partial immunisation |
| Saggurti et al 2018[84] | India | Formation of health-focused self-help groups with women of reproductive age coming from the most marginalised communities.<br>The study evaluates intervention effectiveness and cost-effectiveness. | EAI | DID | 2 | Timeliness |
| Sankar 2013[85] | India | Formation of committees with representatives of the community, local government and service providers to ensure better convergence and coordination of service delivery.<br>The study evaluates intervention effectiveness. | Multiple (EII+EAI) | DID | 30 | FIC, BCG, DPT1, DPT2, DPT3, OPV0, OPV1, OPV2, OPV3, measles, partial immunisation, timeliness |
| Seth et al 2018[86] | India | The study evaluated two different interventions: role of compliance-linked incentives vs text messaging to improve childhood immunisations. Incentive amount was determined after input was received from the local investigators as well as the community workers.<br>The study evaluates intervention effectiveness and cost-effectiveness. | EID | RCT | 9.7 | Partial immunisation, timeliness, attitude about immunisation, attitudes about health providers |
| Shukla 2018[87] | Afghanistan | Community representatives along with health officials identify the health needs of the communities and communicate those to the service providers.<br>The study evaluates intervention effectiveness. | EII | DID | 6 | Supply of CHWs, DPT3 |
| Siddiqi et al 2020[88] | Pakistan | Visual reminders to caregivers in the form of wearable bracelets for the child. The bracelets were designed in consultation with the caregivers.<br>The study evaluates intervention effectiveness. | EID | RCT | 12 | DPT3, measles |
| Tandon et al 1988[89] | India | Enlisting community-based volunteers to motivate and encourage family members to use maternal and child health services.<br>The study evaluates intervention effectiveness. | EAI | CBA | 120 | FIC, BCG, DPT2, DPT3, OPV2 |
| USAID 2008[90] | Ethiopia | Creation of a cadre of community health promoters to carry out behaviour change communication activities in the communities.<br>The study evaluates intervention effectiveness. | EAI | CBA | 48 | FIC, BCG, DPT1, DPT3, OPV3, morbidity, health card availability, measles, dropouts |

Continued

**Table 2** Continued

| Author | Country | Summary of intervention | Engagement type | Study design | Duration (in months) | Outcomes |
|---|---|---|---|---|---|---|
| Webster *et al* 2019[91] | Uganda | Community-based outreach and follow-up with caregivers to improve immunisation uptake and reduce defaulters. The study evaluates intervention effectiveness and cost-effectiveness. | EII | RCT | 12 | Health card availability, BCG, DPT1, DPT2, DPT3, OPV0, OPV1, OPV2, OPV3, IPV, measles, partial immunisation, timeliness, attitude about immunisation, dropouts, morbidity, mortality |
| Younes *et al* 2014[92] | Bangladesh | The intervention involved 162 women's groups who used participatory approaches to discuss maternal and neonatal health issues. The study evaluates intervention effectiveness. | EAI | DID | 20 | FIC, morbidity |

CBA, Controlled before-after; CHWs, community health workers; DID, Difference-in-difference; EAI, Engagement as intervention; EID, Engagement in Design; EII, Engagement in implementation autonomy; FIC, Full immunisation coverage; ITS, Interrupted time series; NGOs, Non-government organisations; RCT, randomised controlled trial.

The true outcomes appear to be heterogeneous ($I^2$=76.8%, $\tau = 0.08$). When low-quality studies were removed, the average effect increased slightly ($\hat{\mu} = 0.11$ (95%CI 0.05 to 0.17), $k = 4$), and was still statistically significant ($z = 3.70$, $p < 0.001$; see online supplemental appendix 9). Publication year was a significant source of heterogeneity; each additional year reduced the size of the effect by .014 SD units (see online supplemental appendix 9).

## Measles

A total of $k = 20$ studies examined the effect on measles vaccination coverage and found a very small but significant benefit for the treated group compared with the untreated group ($\hat{\mu} = 0.07$ (95% CI 0.03 to 0.11), $z = 3.22$, $p < 0.01$; see figure 4). A 95% credibility/prediction interval for the true outcomes is given by −0.08 to 0.22. Hence, although the average outcome

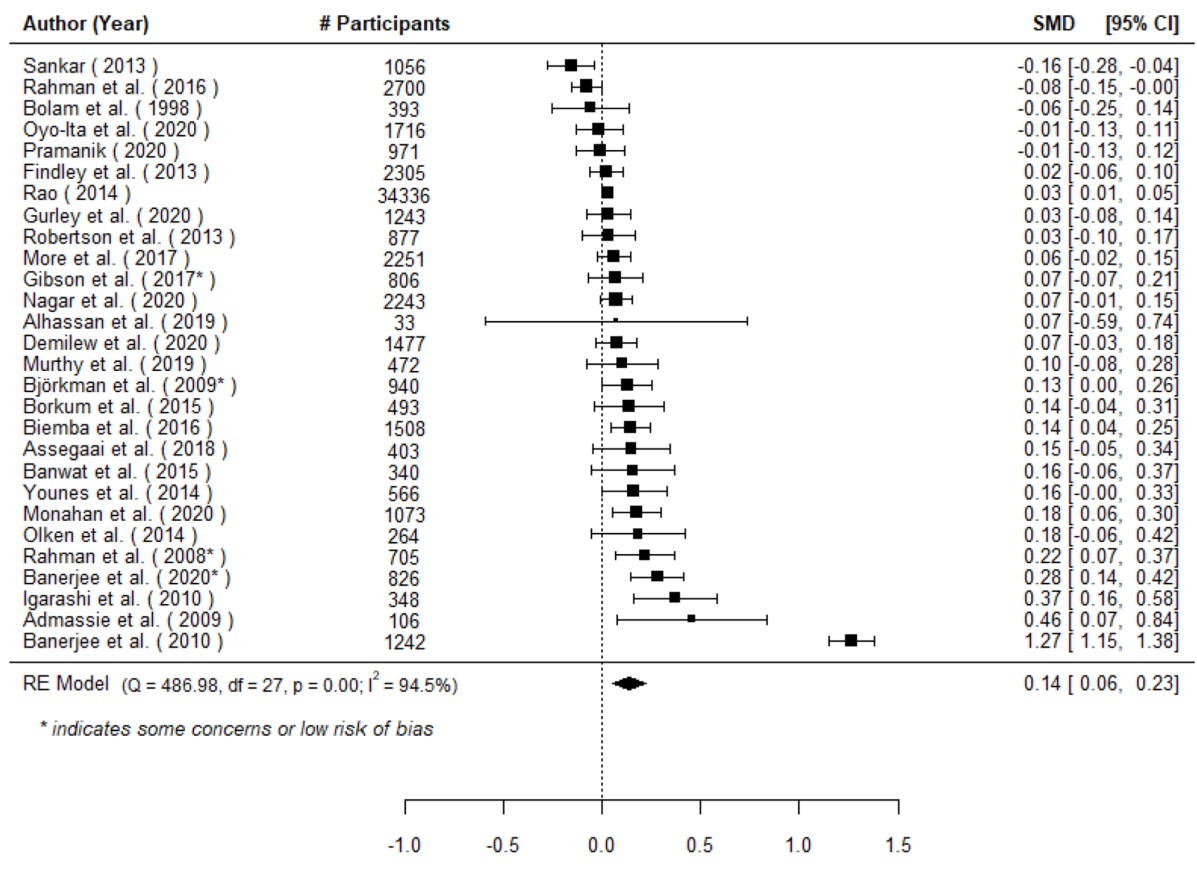

**Figure 2** Forest plot showing the observed outcomes and the estimate of the random effects model for the impact of community engagement interventions on full childhood immunisation. Note that # (number of) participants is specific to each effect and thus may not reflect the sample size for the full study.

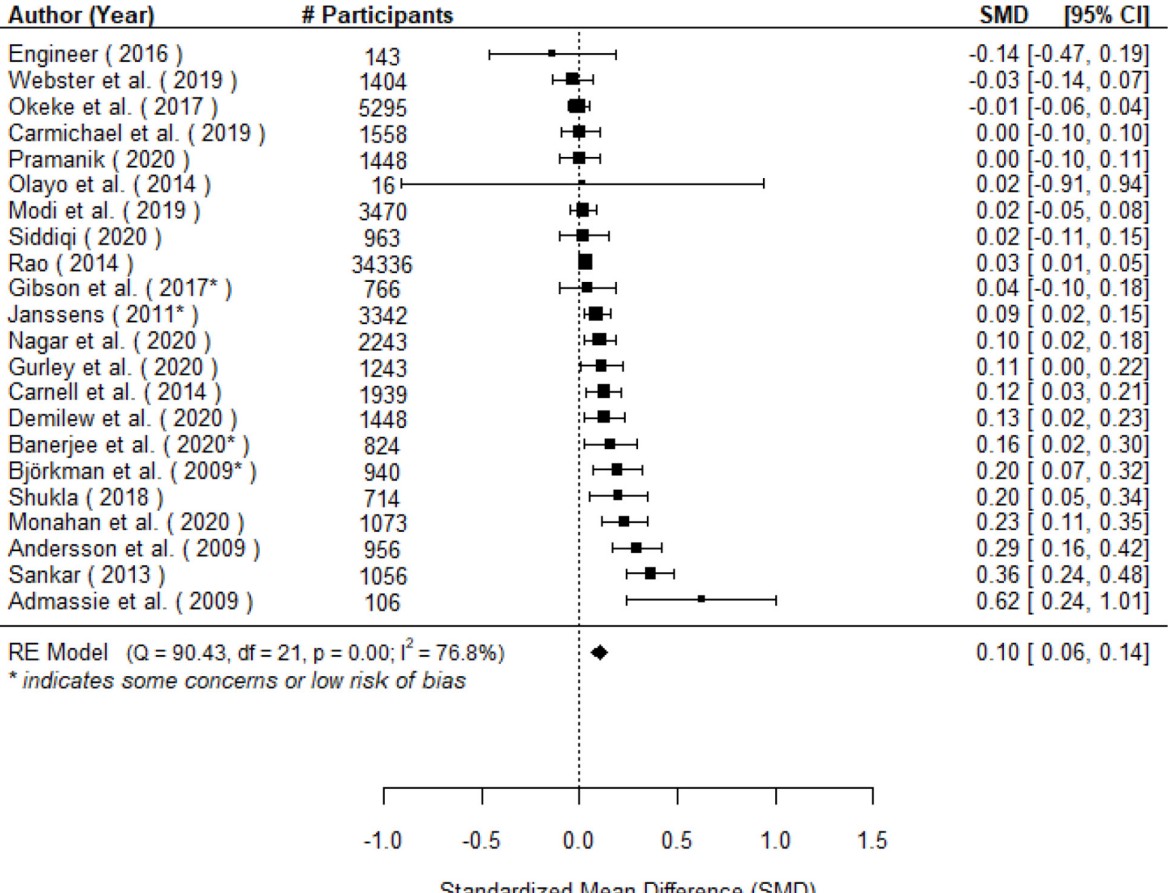

| Author (Year) | # Participants | SMD | [95% CI] |
|---|---|---|---|
| Engineer ( 2016 ) | 143 | -0.14 | [-0.47, 0.19] |
| Webster et al. ( 2019 ) | 1404 | -0.03 | [-0.14, 0.07] |
| Okeke et al. ( 2017 ) | 5295 | -0.01 | [-0.06, 0.04] |
| Carmichael et al. ( 2019 ) | 1558 | 0.00 | [-0.10, 0.10] |
| Pramanik ( 2020 ) | 1448 | 0.00 | [-0.10, 0.11] |
| Olayo et al. ( 2014 ) | 16 | 0.02 | [-0.91, 0.94] |
| Modi et al. ( 2019 ) | 3470 | 0.02 | [-0.05, 0.08] |
| Siddiqi ( 2020 ) | 963 | 0.02 | [-0.11, 0.15] |
| Rao ( 2014 ) | 34336 | 0.03 | [ 0.01, 0.05] |
| Gibson et al. ( 2017* ) | 766 | 0.04 | [-0.10, 0.18] |
| Janssens ( 2011* ) | 3342 | 0.09 | [ 0.02, 0.15] |
| Nagar et al. ( 2020 ) | 2243 | 0.10 | [ 0.02, 0.18] |
| Gurley et al. ( 2020 ) | 1243 | 0.11 | [ 0.00, 0.22] |
| Carnell et al. ( 2014 ) | 1939 | 0.12 | [ 0.03, 0.21] |
| Demilew et al. ( 2020 ) | 1448 | 0.13 | [ 0.02, 0.23] |
| Banerjee et al. ( 2020* ) | 824 | 0.16 | [ 0.02, 0.30] |
| Björkman et al. ( 2009* ) | 940 | 0.20 | [ 0.07, 0.32] |
| Shukla ( 2018 ) | 714 | 0.20 | [ 0.05, 0.34] |
| Monahan et al. ( 2020 ) | 1073 | 0.23 | [ 0.11, 0.35] |
| Andersson et al. ( 2009 ) | 956 | 0.29 | [ 0.16, 0.42] |
| Sankar ( 2013 ) | 1056 | 0.36 | [ 0.24, 0.48] |
| Admassie et al. ( 2009 ) | 106 | 0.62 | [ 0.24, 1.01] |

RE Model (Q = 90.43, df = 21, p = 0.00; I² = 76.8%) 0.10 [ 0.06, 0.14]
* indicates some concerns or low risk of bias

Standardized Mean Difference (SMD)

**Figure 3** Forest plot showing the observed outcomes and the estimate of the random effects model for the impact of community engagement interventions on DPT3 vaccination. Note that # participants is specific to each effect and thus may not reflect the sample size for the full study.

is estimated to be positive, in some studies the true outcome may in fact be negative.

When low-quality studies were removed, the average effect increased ($\hat{\mu}$ = 0.09, k = 6, (95% CI 0.03 to 0.15 and was still statistically significant z=2.98, p=0.003). The true outcomes appear to be heterogeneous (I²=73.6%, $\tau$ = 0.07). None of the moderators were significant sources of heterogeneity (see online supplemental appendix 9).

### Vaccination timeliness

We found a small but significant effect on all three timeliness outcomes: full immunisation timeliness ($\hat{\mu}$ = 0.15 (95% CI 0.07 to 0.24, z = 3.41), p<0.001, 95% prediction interval 0.04 to 0.27; DPT3 timeliness ($\hat{\mu}$ = 0.09 (95% CI 0.03 to 0.14), z = 3.00, p<0.01, 95% prediction interval 0.03 to 0.14) and measles timeliness ($\hat{\mu}$ = 0.23 (95% CI 0.14 to 0.32, z=5.06, p<0.001, 95% prediction interval 0.14 to 0.32. For all timeliness outcomes, tests of heterogeneity were not significant (p>0.05). For full immunisation and measles timeliness outcomes, the sensitivity analysis could not be conducted due to an inadequate number of studies. For DPT3, the average effect increased but became non-significant when low quality studies were removed (see online supplemental appendix 9).

### Subgroups of community engagement interventions

Studies that used engagement as the intervention had a significant positive effect on full childhood immunisation, DPT3 vaccination and measles vaccination but evidence was insufficient to synthesise measures of vaccination timeliness (table 3).

When studies used community engagement in the design, there was a significant positive effect on full childhood immunisation and measles vaccination but not on DPT3 vaccination. In addition, there was a positive significant effect on timeliness of full childhood immunisation and DPT3 vaccination. No studies using engagement in the design reported on timeliness of measles vaccinations.

For engagement in implementation autonomy, the analysis is based on a limited number of studies and we found no significant effect on either coverage or timeliness outcomes. There were no studies reporting on the timeliness of measles vaccination.

Finally, some studies combined multiple engagement types in their interventions. These interventions had a significant effect on DPT3 vaccination but not on measles vaccination or full childhood vaccination. Evidence was insufficient to synthesise measures of vaccination timeliness.

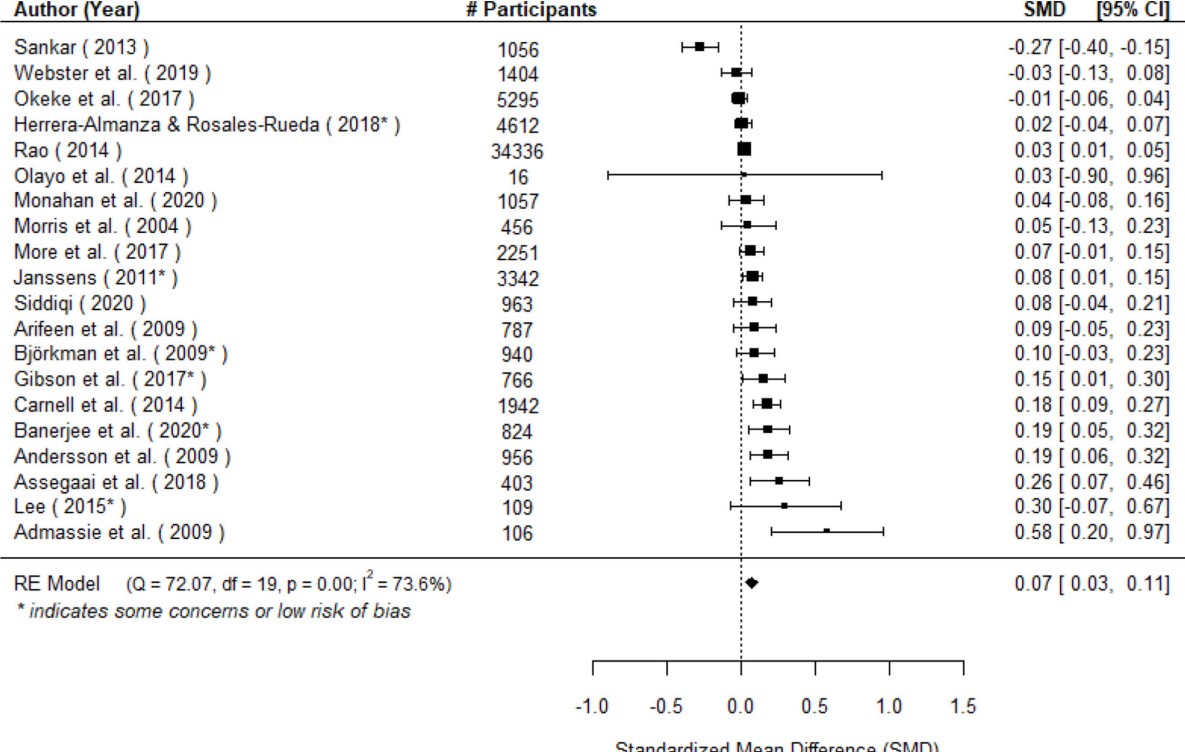

**Figure 4** Forest plot showing the observed outcomes and the estimate of the random effects model for the impact of community engagement interventions on measles vaccination. Note that # participants is specific to each effect and thus may not reflect the sample size for the full study.

Secondary outcomes analyses can be found in online supplemental appendices 10–13.

### Qualitative findings

Programme design characteristics were associated with intervention success or failure across all engagement types. Certain aspects of community engagement itself, such as conducting stakeholder consultations, holding community dialogues or involving community leaders were associated with better immunisation outcomes. Non-engagement intervention design features also affected intervention success. These design features include incentives given to caregivers and leadership and supportive supervision, which improved overall health service delivery and health worker performance. Among the studies that attributed intervention failure to programme design, inadequate duration, frequency or exposure to the intervention were the most notable themes.

The importance of accounting for contextual barriers to or facilitators of immunisation emerged consistently. Limited availability of services, especially insufficient staff and vaccine supply, were dominant barriers to immunisation, affecting outcomes in the early portion of the causal chain. Other common barriers to immunisation included practical barriers faced by caregivers such as costs, largely indirect and logistics (wait time and language barriers) or distance. There was more variation in barriers related to social norms, fear and an understanding of the importance of immunisation by type of engagement. Poor quality of services, including

uninviting attitudes of health workers, posed a barrier to immunisation in communities that received engagement as the intervention or were engaged in the design of the intervention.

However, we also found that certain contextual factors could become facilitators of immunisation outcomes, provided an intervention has adequately situated itself to leverage them. Across all engagement types, studies associated caregivers' awareness and perception of the benefits of vaccination with improved immunisation outcomes. Similarly, availability of health infrastructure and good quality of services were also associated with improved intervention uptake and its impact.

Implementation failures, such as low fidelity, were a common reason for intervention failure. Across all engagement types, interventions did not properly account for existing implementation constraints and practicalities on the ground and were forced to change their implementation plans. Many of these issues were encountered due to uncontrollable factors or invalid theory of change assumptions. For instance, programme design may not have accounted for the unavailability of intended participants due to competing priorities, thereby potentially invalidating the assumption of beneficiary exposure in the casual pathway. Administrative challenges were cited consistently, though their nature varied across engagement types, ranging from technical limitations (such as limited cellphone service) to political constraints to insufficient staffing levels.

**Table 3** Overview of the primary outcomes by engagement type

| | Random effects summary estimate for all interventions g, (95% CI), I², (k) | Engagement as the intervention g, (95% CI), I², (k) | Engagement in the design g, (95% CI), I², (k) | Engagement in the implementation autonomy g, (95% CI), I², (k) | Multiple engagement types g, (95% CI), I², (k) |
|---|---|---|---|---|---|
| Full immunisation | 0.14*, (0.06 to 0.23), 94.46, (28) | 0.08**, (0.03 to 0.13), 70.00, (12) | 010*, (0.02 to 0.19), 23.83, (5) | 0.23, (−0.001 to 0.47), 73.07, (2) | 0.22, (−0.12 to 0.56), 97.94, (9) |
| Measles | 0.07**, (0.03 to 0.11), 73.64, (20) | 0.10***, (0.05 to 0.15), 60.29, (10) | 0.11*, (0.02 to 0.21), 0.00, (2) | 0.03, (−0.09 to 0.15), 54.84, (2) | 0.03, (−0.10 to 0.16), 86.76, (6) |
| DPT3 | 0.10***, (0.06 to 0.14), 76.78, (22) | 0.09**, (0.03 to 0.15), 73.17, (6) | 0.04, (−0.01 to 0.08), 0.00, (6) | 0.11*, (−0.05 to 0.28), 80.12, (3) | 0.20**, (0.06 to 0.34), 87.93, (7) |
| Timeliness (DTP3) | 0.09*, (0.03 to 0.14), 0.00, (7) | N/A | 0.12**, (0.03 to 0.21), 0.00, (4) | 0.04, (−0.06 to 0.13), 0.00, (2) | N/A |
| Timeliness (Measles) | 0.23***, (0.14 to 0.32), 0.00, (3) | N/A | N/A | N/A | N/A |
| Timeliness (full immunisations) | 0.15***, (0.07 to 0.24), 9.66, (5) | N/A | 0.15***, (0.004 to 0.29), 7.202, (2) | 0.38, (−0.28 to 1.03), 0.00, (2) | N/A |

*p<0.05, **p<0.01, ***p<0.001.
NA, not available.

Results were broadly consistent when only qualitative studies with a quality appraisal score greater than 20 were considered. The full qualitative synthesis and sensitivity analysis is available in online supplemental appendix 14.

## Cost-effectiveness findings

Among the 14 studies for which we could calculate cost-effectiveness, we found that the median intervention cost per treated child per vaccine dose (excluding the cost of vaccines) to increase absolute immunisation coverage by one percent was US$3.68 (all costs are reported in 2019 US dollars) and the average cost was US$44.10. There were three outlier observations that drove up this average cost and without them cost per vaccine dose to increase absolute immunisation coverage by 1% averaged US$3.97 (online supplemental appendix 15).

## DISCUSSION
### Principal findings

We found that community engagement interventions had a small, but significant, positive effect on all primary immunisation outcomes related to coverage and their timeliness. We also found that certain features of interventions may contribute to their success. These include (A) appropriate intervention design, including building in community engagement features; (B) addressing common contextual barriers of immunisation and leveraging facilitators and (C) accounting for existing implementation constraints and practicalities on the ground. The median intervention cost per treated child per vaccine dose (excluding the cost of vaccines) to increase absolute immunisation coverage by 1% was US$3.68.

Among the four types of community engagement interventions, we found that engagement as the intervention (embedded community engagement), which involves creation of community buy-in or development of new community-based structures or cadres, had consistent positive effects on more primary vaccination coverage outcomes than the others. We also found engaging the community in the design of the intervention had a positive effect on most primary outcomes related to coverage. We found no ubiquitous patterns of heterogeneity among the primary outcomes.

While zero dose children were not the specific focus of this review, we can offer some insights based on our analyses of both DPT1 and BCG outcomes, which reflect access to initial dosing. Community engagement interventions did not show an effect on DPT1, but the evidence base was of low quality, with six of eight studies assessed as having a high risk of bias. There was a small but significant effect of community engagement interventions on BCG, but here again, the evidence base was of lower quality, with 9 of 12 studies assessed as having a high risk of bias. In both cases, the evidence base was smaller in size than for the primary outcomes. As we find positive effects of community engagement on children returning for DPT and measles doses, it may be that barriers to vaccination,

like availability of health services, for zero dose children are different and unless those are addressed, community engagement itself may not be enough.

### Strengths and weakness

Our systematic review uses a detailed framework of community engagement interventions to assess their effectiveness for improving outcomes related to routine child immunisation in LMICs. As far as we are aware, ours is the first systematic review to do this. Sensitivity analyses excluding high risk of bias studies showed that the effect was slightly larger and still statistically significant for almost all the primary outcomes for which we had sufficient data. The effects were also uniform across geographies and baseline immunisation rates.

We drew on 61 studies for meta-analysis, comprising 31 RCTs and 30 quasi experimental studies. However, only 56 studies provided sufficient information for calculating effect sizes and thus were included in meta-analytical models. For full immunisation, DPT3 and measles coverage, we could draw on 28, 22 and 20 studies, respectively, for pooled effects. However, for the timeliness of these coverage outcomes we had only 0–7 studies to assess the pooled effects. Thus, while for some outcomes the evidence base for drawing conclusions is adequate, for others it is limited. Among the four kinds of community engagement interventions, there was a relatively large evidence base for those with engagement as the intervention and those with multiple engagement types, while for interventions using engagement in implementation autonomy, the evidence was quite limited.

We identified additional documentation comprising qualitative studies, project reports, formative/process evaluations and observation studies for 39 of the 61 included impact evaluations. However, the crucial qualitative papers which help us gain a deeper understanding of overall intervention mechanisms of change were found for only 17 of the 61 IEs. Likewise, only 14 of the primary studies included in this review both estimated the intervention cost and reported it with sufficient detail for the review team to calculate the cost-effectiveness of the treatment. Low-quality cost data and the unavailability of underlying cost data contributed to the small number of primary studies included in the cost-effectiveness analysis.

The quantitative evidence was mostly low quality, though the randomised studies were generally of higher quality and less likely to have confounding bias than the quasi-experimental studies. The quality of qualitative studies was generally high. The quality of the cost evidence was mixed. Despite the quality concerns about quantitative evidence, the sensitivity analysis conducted by excluding low-quality studies corroborated the overall findings. Despite a comprehensive search strategy and the inclusion of grey literature, publication bias was detected for the three primary coverage outcomes (full immunisation, DPT3 and measles). While bias correction analyses indicated an identical effect size for full immunisation and DPT3, the effect for measles was reduced when publication bias was corrected for. Timeliness outcomes had an insufficient number of studies to test for publication bias, which limits our ability to interpret heterogeneity.

### Agreement and disagreement with other reviews

The findings from this review are broadly consistent with Molina *et al*,[5] which found positive effects of community monitoring interventions on immunisation coverage. Another review by Gilmore and McAuliffe[4] examined the effectiveness of preventive interventions delivered by community health workers for maternal and child health in LMICs on essential newborn care and found some evidence in its support through narrative synthesis, but found the evidence base to be insufficient to draw firm conclusions.

### Limitations

There are several potential limitations to the current review: (A) there were few analyses that were sufficiently powered to test for publication bias, thus,we cannot rule this out in many cases; (B) many of the moderator analyses were underpowered, meaning that in many cases we were unable to explore heterogeneity. This was particularly true in the context of the subgroup analyses of the four intervention types. In addition, it is likely that there is interdependency among moderator variables, but the current study did not allow for us to disentangle these confounds. Future studies may aim to better assess how moderators may work in tandem to affect the magnitude of change; (C) even in cases where the average effect was significant, forest plots demonstrate that some of the included studies reported a small negative affect, and prediction intervals often included both positive and negative values, which may have important implications when making decisions related to programme design and implementation; (D) we also observed very few studies which focused on subpopulation groups. This is particularly problematic given the focus on LMICs, where equity is important to consider when trying to increase coverage; (E) most of the community engagement interventions were in combination with other intervention components, thus we were not able to establish their unique contribution to changes in outcomes and (F) inclusion of primary studies into this review was based on the description of the community engagement aspects of the intervention. We may have excluded studies that should ideally have been included because of inadequate reporting of intervention components. Finally, funding for this project has concluded, thus, we do not have the resources to update our literature search last conducted in May of 2020.

### Implications for policy and practice

COVID-19 has impacted routine child immunisation negatively in some countries, and community engagement interventions could be an effective way to counteract this decline. The positive effects of community engagement interventions can be expected across a variety

of settings, although some engagement approaches appear to be more effective than others. Positive design features should be integrated into these interventions, including features such as holding community dialogues or involving community leaders, and non-community engagement features such as local supportive supervision and incentives to healthcare workers or caregivers. Wherever possible, binding contextual barriers to immunisation, such as weak health systems and social norms, should be accounted for in the design of interventions. Existing contextual facilitators for immunisation, such as good existing health systems or high maternal education, could be leveraged for increasing intervention impacts. Important implementation preconditions, such as regular internet service or sufficient staffing, should be assessed and established before the implementation or addressed through the design itself. Close monitoring of intervention implementation along with good understanding of context is important to help make necessary modifications in case of unexpected challenges, such as political instability.

## Further research

For better-quality evidence and deeper mechanistic understanding, policy makers and practitioners should consider prioritising funding or commissioning research in the following areas: (A) ways of ameliorating outcome measurement bias due to self-reported immunisation coverage outcomes, as this was a principal source of bias; (B) better reporting of interventions, more rounded analysis of why the interventions worked through mixed-methods evaluations and greater focus on intermediate outcomes for improved understanding of causal mechanisms; (C) collection and reporting of high-quality cost data to enable cost-effectiveness analysis, which is important for decision-making within budget constraints and (D) focus on subgroup analysis, including for zero dose children, for ensuring immunisation services for the most marginalised children. It would also be useful to conduct an update of this review to include evidence produced since our final literature search in May of 2020.

**Acknowledgements** This research has been undertaken as a part of 3ie's immunisation evidence programme, supported by the Bill and Melinda Gates Foundation, Seattle, USA. We would like to thank Sohail Agha, Senior Program Officer, Gates Foundation, Seattle, for his continued support and engagement on the evidence programme and the review. Special thanks to Molly Abbruzzese whose guidance helped shape the scope of our evidence programme and aided the conceptualisation of this review. We thank Lisa Menning, Danielle Pedi, Jennifer (Fluder) Siler and Pedja Stocjicic for their valuable feedback on the review. We wish to thank John Eyers, Yoav Vardy, Daniela Anda, Shradha Parsekar, Sejal Luthra, Yue Zhan, Aditi Hombali, Lalitha Vadrevu, Reva Datar, Pankhuri Jha, Beáta Berkovics, Ashton Baafi, Meital Kupfer David Atika, Himani Aggarwal, Agrima Sahore, Mansi Wadhwa, and Harini Narayanan for excellent support and research assistance. We would further like to thank Marie Gaarder, Executive Director, Birte Snilstveit, Director of Synthesis and Reviews Office and Sebastian Martinez, Director of Evaluation Office, 3ie, for their guidance on the review process.

**Contributors** MJ, ME and AB conceived the review and wrote the initial protocol. ME and AB did the systematic search. ME, MJ and AB screened and identified studies and MJ and AB made final decisions regarding study inclusion. SS did the statistical analysis. CL and AB did the qualitative analysis. EB synthesised the cost evidence. MJ provided critical inputs on the whole analysis, checked data, coordinated the review and had full access to all materials and results. External consultants supported the authors in search, screening, data extraction and critical appraisal of quantitative and qualitative evidence base. All authors critically reviewed and revised the manuscript and approved the final document for submission. MJ is responsible for the overall content and is the guarantor.

**Funding** This research was funded through a grant from the Bill & Melinda Gates Foundation (OPP1115129).

**Disclaimer** The funder of the study had no role in study design, data collection, data analysis, data interpretation, or writing of the report.

**Competing interests** The International Initiative for Impact Evaluation (3ie). Through this grant, 3ie provided funding and technical assistance for seven impact evaluations of community engagement interventions for immunisation as a part of its immunisation evidence programme. This technical assistance included, but was not limited to: reviewing study designs, analysis plans and data collection instruments; advising research teams on how to improve study components and address challenges that arise during the course of the evaluation; and supporting grantees in engaging with stakeholders to promote uptake and use of evidence generated by the evaluations. As members of 3ie staff, authors MJ, AB and ME have all had varying levels of involvement in reviewing deliverables for these evaluations and providing research teams with technical assistance. Several procedural safeguards and transparency measures were put in place to mitigate the risk this conflict of interest imposed. First, all candidate studies, including those funded by 3ie, underwent a rigorous multi-step screening process, including review at the title, abstract, and full-text levels. To qualify for inclusion in the SR, a study was judged to meet the inclusion criteria related to study design, outcomes and population by two independent screeners who have reviewed the full text of the study. The 3ie study authors were responsible for assessing whether the studies met the inclusion criteria for community engagement because of the complexity of the framework. However, these authors have no financial interest in this area and have not published any prior reviews on the topic. The remaining study authors have no conflicts of interest to declare.

**Patient and public involvement** Patients and/or the public were not involved in the design, or conduct, or reporting, or dissemination plans of this research.

**Patient consent for publication** Not applicable.

**Ethics approval** Ethical consent from an ethics committee or institutional board was not required as this study does not involve human subjects.

**Provenance and peer review** Not commissioned; externally peer reviewed.

**Data availability statement** Data are available on reasonable request.

**ORCID iDs**
Monica Jain http://orcid.org/0000-0001-5428-377X
Charlotte Lane http://orcid.org/0000-0002-7730-6687
Mark Engelbert http://orcid.org/0000-0002-3665-1257

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
