## [Reviewer comments · BMJ Open]

ARTICLE DETAILS

TITLE (PROVISIONAL)	Use of community engagement interventions to improve child immunisation in low- and middle-income countries: a systematic review and meta-analysis
AUTHORS	Jain, Monica; Shisler, Shannon; Lane, Charlotte; Bagai, Avantika; Brown, Elizabeth; Engelbert, Mark

VERSION 1 – REVIEW

REVIEWER	Rothstein, Hannah The City University of New York, Baruch College Management
REVIEW RETURNED	31-Mar-2022

GENERAL COMMENTS	This paper is interesting and well written, and it has the potential to make a contribution to public health. I had three main statistical concerns which threaten the results and conclusions of the study. The first is that the authors have not addressed the issue of heterogeneity sufficiently. Although, by their own report, there is a lot of heterogeneity in effect sizes, it is essentially ignored. The authors do use it as a basis for examining some moderators, but that is not the key issue here. The issue is that, given the amount of heterogeneity present, the authors should not be focusing on the mean effect, but rather the dispersion in effects. In this case, the mean is actually misleading. As the forest plot shows, there are some observed effects that are negative and some that are positive. If the authors had calculated a prediction interval, they would have seen that the range of the population of effects crosses the zero line. In other words, some of the interventions had a negative effect on the outcome. This has important implications. The authors need to report tau in all cases, not only when it is estimated to be zero, and must report and then call attention to the width of the prediction interval, emphasizing that it includes both positive and negative effects. Secondly, the authors don't clearly distinguish between moderators that were hypothesized a priori and those moderator analyses that were post hoc and exploratory. Given the number of post hoc moderator analyses, it is likely that some of the significant moderators arose by chance. Third, the authors fail to point out the likely confounding of moderators. It is highly unlikely that they are independent of each other. Although they cannot disentangle these confounds in the current study, they could examine how often the moderators co-occur within each of the primary studies. I also have some minor concerns: 1. Why do the authors use the rank correlation test for publication bias when the regression test is more powerful? (I have concerns about the appropriateness of publication bias analyses when there
---

	is a lot of heterogeneity, but unfortunately it is common practice and I will not fault the current authors for conducting them) 2. To show that none of the studies are overly influential, a one-left-out analysis is more accessible and audience-friendly. 3. If the authors have no reason to suspect that the interventions work differently for DPT than for Measles, why are they conducting separate analyses for each type of vaccine? I suggest that they consider creating a composite variable. I hope the authors take these considerations to heart and produce a revision that is ready for publication.
--	--

REVIEWER	Dadari, Ibrahim University of South Florida
REVIEW RETURNED	18-Apr-2022

GENERAL COMMENTS	This paper titled "Use of community engagement interventions to improve child immunization in low- and middle-income countries: a systematic review and meta-analysis" is quite timely and will contribute significantly to evidence in community engagement for improving routine childhood vaccination. Few observations below to make the paper stronger and the message clearer: Page 5 Line 51 – 53: “The grey literature search was conducted by study authors and external consultant reviewers” Here it will be good to indicate the initials of authors who did the search and or review ... Line 56: definition of “community” for this study should be defined in the introduction section of the paper. The authors should also mention the definition used for RCT and quasi-experimental studies in this review The inclusion and exclusion criteria for the papers should be clearly defined. For now, it seems to be mixed up and less structured. Perhaps listing the inclusion and exclusion criteria under separate sub-headings could be useful. Page 6 Line 18 – 21: “.....all papers were double screened”. Specify the initials of who and who double-screened the papers. Also, specify who did the reconciliation Line 24: “qualitative papers and project documents were screened....”. The statement isn’t complete, please detail whether papers were screened in or out, how that was accomplished, and by who - again initials could be useful here? Also briefly mention in 1 or 2 sentences how the coding was done. Line 28: Full immunization – specify what full immunization means while reviewing studies done across different countries with different immunization schedules Line 29: “measles” – did the authors include studies with 1st or 2nd dose of measles-containing vaccine or both? What was considered timely vs untimely doses in this study – should be explained clearly? Page 8
---

	Line 5: briefly justify why you used a random effect model – merits and risks in this study. Also, strengthen the limitations section of the paper. NB. This paper would have been even greater if one of the outcome measures included “zero-dose” children that is to have included DPT1 vaccine in the analysis; particularly also the cost component. As part of the Immunization Agenda 2030, there is an increased focus on reaching zero-dose children which serve as a marker of multiple deprivations.
--	--

REVIEWER	Di Ruggiero, Erica University of Toronto, Dalla Lana School of Public Health
REVIEW RETURNED	25-Apr-2022

GENERAL COMMENTS	The authors address a clear and important objective, that is to examine the effectiveness and cost-effectiveness of community engagement interventions on routine childhood immunisation in LMICs, including contextual, design, and implementation features associated with effectiveness of these interventions. Overall, the manuscript is clearly written and summarizes the results of a comprehensive review. One of the notable strengths is the framework to nuance different types of community engagement interventions. The following comments are intended to help strengthen the manuscript Introduction:  - could better articulate what constitutes an 'immunization' intervention - The authors state "there is a dearth of rigorous and systematic evidence about whether and how community engagement interventions work to increase routine childhood immunisations, for whom, or at what cost" - Suggest you add and in which contexts (given interest in LMICs); second, there is no reference or evidence to substantiate this statement, and why focus on 'childhood immunisations' What research questions guided your review? What the PICO strategy used or modified to construct your question(s)? Methods - suggest you clarify upfront that this a systematic review of academic and grey literature or are both actually included? (There is a reference to grey literature in the second paragraph of this section). On a related note, were the same search strategies used for both academic and grey literature? This isn't clear in the main text of the manuscript. what are the inter-rater reliability between the 2 reviewers who coded? Results What is the breakdown of included papers (by language) or should we assume all were in English? on what basis did you assign papers to more than one engagement type? Table 1 - it would have been helpful to capture the research question or purpose for each included study in this table to illustrate studies that were about effectiveness vs. cost-effectiveness.
---

	Did articles included in your review capture how long the immunisation program had been running in the country, as this could impact coverage and timeliness? Second, funding sources (government, GAVI, other sources or combination thereof) could have also affected effectiveness. Are you able to comment based on your review? Qualitative findings - based on your results, it appears as if studies reported context (including barriers and facilitators) in a variety of ways. Based on the studies included, was context explicitly considered/mentioned in individual study questions? How did this vary based on type of community engagement approach reported? When reporting intervention design, it is not always clear who designed the intervention and whether those involved in design were different from implementation? Cost-effectiveness - how and when were costs captured in included studies for which you could assess cost-effectiveness of interventions? (at design, implementation stage, unknown?) Some additional details would be helpful Discussion "The quantitative evidence was mostly low quality, though the randomised studies were generally of higher quality than quasi-experimental studies" - how so? Needs elaboration. I was surprised that there was no discussion about differential impacts of interventions on sub-population groups as equity is an important consideration in coverage and in highlighting who is being left behind (perhaps this is a limitation of the studies included, which should be discussed, given focus on LMICs and an area for further investigation) There is no discussion of the limitations of the study included in the discussion section (including limitations or challenges with applying your community engagement framework, in addition to the nature of the studies included)
--	---

REVIEWER	Wariri, Oghenebrume LSHTM
REVIEW RETURNED	27-Apr-2022

GENERAL COMMENTS	In this manuscript, Lane et al. systematically examined the effectiveness and cost-effectiveness of community engagement interventions on routine childhood immunisation outcomes in low-and middle-income countries and identified contextual, design, and implementation features associated with effectiveness. This is a very detailed, and well-written manuscript. The authors clearly documented all the processes involved in this systematic review and meta-analysis. This is evidenced by the number of documents which were prepared by the authors and attached as supplementary materials. That being said, I have the following few comments that the authors can consider addressing to improve the overall quality of their manuscript. I have listed them below  1. Throughout the manuscript the authors used L&MICs. The authors should consider writing this as LMICs.
---

	2. On page 4, lines 19-21, I do not think the authors should make this about COVID-19. The authors have a legitimate and very important research question and do not need to pivot to COVID-19 when the sentence was supposed to be about how community engagement impacts routine childhood vaccination. 3. The section on page 4, lines 38-53, and page 5, lines 3-22. I believe this whole section might be better if it is moved to the methods section, under the subheading 'conceptual framework'. This would then mean that the authors need to rework the introduction section substantially as it will be reduced to just two paragraphs. In its current form, the purpose of this section within the introduction appears unclear and also looks disjointed from the first two paragraphs. This is even made worse by the fact that the authors have already stated their aim and objectives and one would naturally expect that the next section after the aim and objectives will be the methods section. 4. On page 5, lines 47-48, It will be important for the authors to put a time stamp. That is, World Bank country income classification as of when? This is important because country income status changes and could determine which countries would have been included in the review. 5. Page 7, lines 15-35 should be under a separate subheading 'Risk of bias assessment'. Relatedly, the authors should consider including a colour-coded, risk of bias assessment figure that clearly rates the included studies. 6. On page 8, lines 8-13, the authors should clarify if these themes were determined a priori
--	---

VERSION 1 – AUTHOR RESPONSE

The first is that the authors have not addressed the issue of heterogeneity sufficiently. Although, by their own report, there is a lot of heterogeneity in effect sizes, it is essentially ignored. The authors do use it as a basis for examining some moderators, but that is not the key issue here. The issue is that, given the amount of heterogeneity present, the authors should not be focusing on the mean effect, but rather the dispersion in effects. In this case, the mean is actually misleading. As the forest plot shows, there are some observed effects that are negative and some that are positive. If the authors had calculated a prediction interval, they would have seen that the range of the population of effects crosses the zero line. In other words, some of the interventions had a negative effect on the outcome. This has important implications. The authors need to report tau in all cases, not only when it is estimated to be zero, and must report and then	Prof. Hannah Rothstein, The City University of New York	Quant	We thank the reviewer for this thoughtful comment. While we did calculate prediction intervals, we did not include them in text as the estimate of the prediction interval will be imprecise if the estimates of the summary effect are imprecise, for example, if they are based on only a few studies. Given that the vast majority of our analyses included few studies, we thought it was best not to include the prediction intervals in this case. As requested, we did add tau to the main body of the article for all of the primary analyses. For each of the analyses presented in the appendices, tau squared was already presented. We agree that it is clear from the forest plots that in some cases, small negative effects are observed. We have now highlighted this issue in the limitations section.
---	--	--------------	--

call attention to the width of the prediction interval, emphasizing that it includes both positive and negative effects.			
Secondly, the authors don't clearly distinguish between moderators that were hypothesized a priori and those moderator analyses that were post hoc and exploratory. Given the number of post hoc moderator analyses, it is likely that some of the significant moderators arose by chance.	Prof. Hannah Rothstein, The City University of New York	Quant	All but two moderators were selected a priori. The moderators for baseline coverage and vaccine hesitancy were added after feedback from an initial peer review from Campbell (co-publisher of this work). A footnote to this effect has been added to clarify.
Third, the authors fail to point out the likely confounding of moderators. It is highly unlikely that they are independent of each other. Although they cannot disentangle these confounds in the current study, they could examine how often the moderators co-occur within each of the primary studies.	Prof. Hannah Rothstein, The City University of New York	Quant	This is an interesting point, and we have added a note about the likelihood of dependence to the discussion.
1. Why do the authors use the rank correlation test for publication bias when the regression test is more powerful? (I have concerns about the appropriateness of publication bias analyses when there is a lot of heterogeneity, but unfortunately it is common practice and I will not fault the current authors for conducting them)	Prof. Hannah Rothstein, The City University of New York	Quant	We are slightly confused by this comment, as we have reported both the regression test and the rank correlation test (see for example p. 27 in the clean manuscript).
2. To show that none of the studies are overly influential, a one-left-out analysis is more accessible and audience-friendly.	Prof. Hannah Rothstein, The City University of New York	Quant	We utilized a left-one-out analysis any time that a potential outlier was indicated by the Cooks distance or studentized residuals. We have clarified this in the methods section.

3. If the authors have no reason to suspect that the interventions work differently for DPT than for Measles, why are they conducting separate analyses for each type of vaccine? I suggest that they consider creating a composite variable.	Prof. Hannah Rothstein, The City University of New York	Quant	As per the routine immunization schedule in most L&MICs, measles vaccines are administered at nine months. Whereas, DPT3 and other vaccinations are typically administered by 3-5 months of age. Given the gap between the vaccinations, we were interested in understanding whether immunization interventions differentially impact measles vaccination uptake and help prevent drop-outs. We also referred to the guidelines issued by WHO which recommend that DTP1 to DTP3, BCG to the measles-containing virus (MCV1), and MCV1 to MCV2 should be used as indicators of immunization dropout (https://cdn.who.int/media/docs/default-source/documents/ddi/facilityanalysisguide-immunization.pdf?sfvrsn=3cb62a74_2&download=true).
Line 51 – 53: “The grey literature search was conducted by study authors and external consultant reviewers” Here it will be good to indicate the initials of authors who did the search and or review	Dr. Ibrahim Dadari, University of South Florida	Methods	This has been clarified in the manuscript.
Line 56: definition of “community” for this study should be defined in the introduction section of the paper. The authors should also mention the definition used for RCT and quasi-experimental studies in this review	Dr. Ibrahim Dadari, University of South Florida	Methods	The definition of "community" has been added in Methods section under Conceptual Framework and of RCT and quasi-experimental studies under Inclusion/Exclusion criteria (PICOS)
The inclusion and exclusion criteria for the papers should be clearly defined. For now, it seems to be mixed up and less structured. Perhaps listing the inclusion and exclusion criteria under separate sub-headings could be useful.	Dr. Ibrahim Dadari, University of South Florida	Methods	We appreciate the advice on additional clarity. A table reflecting the PICOS is now included.
Line 18 – 21: “.....all papers were double screened”. Specify the initials of who and who double-screened the papers. Also, specify who did the reconciliation	Dr. Ibrahim Dadari, University of South Florida	Methods	This has been clarified in the manuscript.
Line 24: “qualitative papers and project documents were screened...”. The statement isn’t complete, please detail whether papers were screened in or out, how that was accomplished, and by who - again initials could be useful here? Also briefly mention in 1 or 2 sentences how the coding was done.	Dr. Ibrahim Dadari, University of South Florida	Methods	Information on who conducted screening and how is in the end of the "Screening" section. A description of how coding was done is in the last paragraph of "Data analysisis."
Line 28: Full immunization – specify what full immunization means while reviewing studies done across different countries with different immunization schedules	Dr. Ibrahim Dadari, University of South Florida	Methods	We use the WHO's definition of full immunization for this study, which is "the percentage of one-year-olds who have received one dose of Bacille Calmette-Guérin (BCG) vaccine, three doses of polio vaccine, three doses of the combined diphtheria, tetanus toxoid and pertussis (DTP3) vaccine, and one dose of measles vaccine"

			This has been clarified in the manuscript.
Line 29: "measles" – did the authors include studies with 1st or 2nd dose of measles-containing vaccine or both? What was considered timely vs untimely doses in this study – should be explained clearly?	Dr. Ibrahim Dadari, University of South Florida	Methods	We included the uptake of first dose of measles as an outcome indicator for this study. This has been clarified in the manuscript.
Line 5: briefly justify why you used a random effect model – merits and risks in this study.	Dr. Ibrahim Dadari, University of South Florida	Methods	We have added text to this section reflecting our decision process. Fixed effects models are only appropriate when you can reasonably expect that all of the included studies are functionally identical (e.g. participants are all drawn from the same pool and researchers use the same interventions, measures, etc.). We were aware from our initial readings that this condition was not met in our group of studies, indicating a random effects model was most appropriate. In addition, the fixed effects model computes the effect size only for the pool of participants from which the researchers are drawing, and does not generalize to other populations. Again, this was not the goal of the current analysis, and thus the random effect model was deemed most appropriate. (See Bornstein et al., 2021 for a full discussion).
strengthen the limitations section of the paper.	Dr. Ibrahim Dadari, University of South Florida	Methods	We have added a paragraph in the discussion section that is dedicated to the discussion of study limitations.
NB. This paper would have been even greater if one of the outcome measures included "zero-dose" children that is to have included DPT1 vaccine in the analysis; particularly also the cost component. As part of the Immunization Agenda 2030, there is an increased focus on reaching zero-dose children which serve as a marker of multiple deprivations	Dr. Ibrahim Dadari, University of South Florida	Methods	We thank the reviewer for this important suggestion. We have now added a paragraph in the discussion section on findings for DPT1 and BCG outcomes in relation to the zero dose children
could better articulate what constitutes an 'immunization' intervention	Dr. Erica Di Ruggiero, University of Toronto	Introduction	Thanks for pointing out the use of term "intervention" which is not the best given later we are talking out community engagement interventinos. We have replaced this term with "ways" and is also consistent with how GAVI described this here - https://www.gavi.org/vaccineswork/value-vaccination/cost-effective

The authors state "there is a dearth of rigorous and systematic evidence about whether and how community engagement interventions work to increase routine childhood immunisations, for whom, or at what cost" - Suggest you add and in which contexts (given interest in LMICs); second, there is no reference or evidence to substantiate this statement, and why focus on 'childhood immunisations'	Dr. Erica Di Ruggiero, University of Toronto	Introduction	We have made it explicit that the literature on community engagement is lacking for LMICs, We have also provided references to the existing sparse evidence in the form of systematic reviews. We have focused on childhood immunization because of the sheer size of the unvaccinated children in LMICS as has been mentioned in the first paragraph in the introduction
What research questions guided your review? What the PICO strategy used or modified to construct your question(s)?	Dr. Erica Di Ruggiero, University of Toronto	Introduction	We have added the research questions that guided our review under the Methods section. We have also added the PCO strategy used for our review under the Methods section. PICO was not modified during the review
Methods - suggest you clarify upfront that this a systematic review of academic and grey literature or are both actually included? (There is a reference to grey literature in the second paragraph of this section). On a related note, were the same search strategies used for both academic and grey literature? This isn't clear in the main text of the manuscript.	Dr. Erica Di Ruggiero, University of Toronto	Methods	We have provided additional description in this paragraph of how the grey literature search was conducted, and clarified that search strings were developed on a site-by-site basis for grey literature websites.
what are the inter-rater reliability between the 2 reviewers who coded?	Dr. Erica Di Ruggiero, University of Toronto	Methods	We have now included information on reliabilities for a subset of studies. All studies were double coded and reconciled, with a third core team member resolving any disagreements.
What is the breakdown of included papers (by language) or should we assume all were in English?	Dr. Erica Di Ruggiero, University of Toronto	Results	We added a sentence clarifying that we found one Spanish publication, with all others in English.
on what basis did you assign papers to more than one engagement type?	Dr. Erica Di Ruggiero, University of Toronto	Results	This is now clarified at the beginning of the data analysis section.
Table 1 - it would have been helpful to capture the research question or purpose for each included study in this table to illustrate studies that were about effectiveness vs. cost-effectiveness.	Dr. Erica Di Ruggiero, University of Toronto	Results	The primary objective of the studies included in this review is to measure intervention effectiveness. A sub-set of these evaluations also reports some form of a cost analysis. The intervention descriptions in Table 1 have been amended to include information on whether a study only measured intervention effectiveness or it measured both effectiveness and cost-effectiveness.

Did articles included in your review capture how long the immunisation program had been running in the country, as this could impact coverage and timeliness?	Dr. Erica Di Ruggiero, University of Toronto	Results	Unfortunately this data was not extracted as part of this project, though we did collect the time of exposure to the intervention of the included participants, and have used this in tests of moderation.
Second, funding sources (government, GAVI, other sources or combination thereof) could have also affected effectiveness. Are you able to comment based on your review?	Dr. Erica Di Ruggiero, University of Toronto	Results	We looked at implementers rather than funders, as we saw this as the more policy-relevant question. We included this as a potential moderator of intervention effectiveness. However, based on the data extracted for this review (which did not include funder names) we cannot comment on the impact of specific funding sources. We will certainly keep this in mind should be update this review in the future.
Qualitative findings - based on your results, it appears as if studies reported context (including barriers and facilitators) in a variety of ways. Based on the studies included, was context explicitly considered/mentioned in individual study questions? How did this vary based on type of community engagement approach reported?	Dr. Erica Di Ruggiero, University of Toronto	Qual	This is an interesting question. Unfortunately, we do not have the data to address it. We did not extract information regarding the study questions of individual evaluations and are not able to conduct additional data extraction at this time.
When reporting intervention design, it is not always clear who designed the intervention and whether those involved in design were different from implementation?	Dr. Erica Di Ruggiero, University of Toronto	Qual	We are not totally clear on this question, and apologize if we do not answer adequately. If this question regards the actual organizations designing and implementing each intervention, that seems to be too much detail to provide in this systematic review publication, which is relatively short. We have a more extensive report to be published with Campbell, but need to make strategic decisions regarding the information to present here. It is not clear that readers will be interested in the designers and implementers of each intervention.

Cost-effectiveness - how and when were costs captured in included studies for which you could assess cost-effectiveness of interventions? (at design, implementation stage, unknown?) Some additional details would be helpful	Dr. Erica Di Ruggiero, University of Toronto	Cost	It most cases, it is difficult to tell how and when costs were captured. Overall we find that indications of the quality of underlying cost data are mixed. Eight of 21 evaluations used expenditure reports to generate cost estimates, two used budgets and the remaining 11 evaluations provided no information on the provenience of the underlying cost data that was used in the analysis. About half of the included evaluations appear to have carried out a planned, organized, cost analysis. Twelve of 21 evaluations clearly state the form of economic evaluation (i.e. cost-effectiveness analysis); nine studies report the perspective of the costing, which is key for judging the correct inclusion and exclusion criteria for the components of a total cost estimate; and ten of 21 evaluations describe the method of costing that was used to collect cost data (i.e. the ingredients method). Last, we find that the quality of the descriptive detail on reported costs was mixed. Just over half of all evaluations (13 of 21) provided thorough, descriptive information on costs; two evaluations provided some descriptive information, e.g. a breakdown of key unit costs; and six evaluations gave very minimal or no descriptive information on costs. Details on cost analysis are in the extensive report to be published by Campbell. Because of word limit of manuscript, it is not clear if this much detail on cost analysis would be useful for the reader.
"The quantitative evidence was mostly low quality, though the randomised studies were generally of higher quality than quasi-experimental studies" - how so? Needs elaboration.	Dr. Erica Di Ruggiero, University of Toronto	Discussion	The randomized studies were less likely to have confounding bias than the quasi-experimental studies and this has been clarified in the discussion section
I was surprised that there was no discussion about differential impacts of interventions on sub-population groups as equity is an important consideration in coverage and in highlighting who is being left behind (perhaps this is a limitation of the studies included, which should be discussed, given focus on LMICs and an area for further investigation)	Dr. Erica Di Ruggiero, University of Toronto	Discussion	It's a great point, and we were equally surprised at the lack of sub-group analyses presented in the included studies (that were reported in a way that allowed for effect size calculations). As you suggest, we now highlight this in the limitations section.
There is no discussion of the limitations of the study included in the discussion section (including limitations or challenges with applying your community	Dr. Erica Di Ruggiero, University of Toronto	Discussion	We have added a paragraph in the discussion section that is dedicated to the discussion of study limitations, including on nature of community engagement

engagement framework, in addition to the nature of the studies included)			interventions and challenges of applying community engagement framework
1. Throughout the manuscript the authors used L&MICs. The authors should consider writing this as LMICs.	Dr. Oghenebrume Wariri, LSHTM		We have replaced L&MICs with LMICs
2. On page 4, lines 19-21, I do not think the authors should make this about COVID-19. The authors have a legitimate and very important research question and do not need to pivot to COVID-19 when the sentence was supposed to be about how community engagement impacts routine childhood vaccination	Dr. Oghenebrume Wariri, LSHTM	Introduction	Good point! We have taken off the reference to Covid in the introduction
3. The section on page 4, lines 38-53, and page 5, lines 3-22. I believe this whole section might be better if it is moved to the methods section, under the subheading 'conceptual framework'. This would then mean that the authors need to rework the introduction section substantially as it will be reduced to just two paragraphs. In its current form, the purpose of this section within the introduction appears unclear and also looks disjointed from the first two paragraphs. This is even made worse by the fact that the authors have already stated their aim and objectives and one would naturally expect that the next section after the aim and objectives will be the methods section.	Dr. Oghenebrume Wariri, LSHTM	Introduction	The whole section on community engagement has been moved to the Methods section as suggested by the reviewer. We have also reworked the introduction substantially.
4. On page 5, lines 47-48, It will be important for the authors to put a time stamp. That is, World Bank country income classification as of when? This is important because country income status changes and could determine which countries would have been included in the review.	Dr. Oghenebrume Wariri, LSHTM	Methods	We have updated the language to clarify that we used countries' income status at the time the intervention began to determine L&MIC status.
5. Page 7, lines 15-35 should be under a separate subheading 'Risk of bias assessment'. Relatedly, the authors should consider including a colour-coded, risk of bias assessment figure that clearly rates the included studies.	Dr. Oghenebrume Wariri, LSHTM	Results	We have added the subheading as the reviewer suggests. The figures the reviewer refers to are now presented in Appendix 8. We have also made this more explicit for the reader.
6. On page 8, lines 8-13, the authors should clarify if these themes were determined a priori	Dr. Oghenebrume Wariri, LSHTM	Qual	This has been clarified at the end of the data analysis section.

VERSION 2 – REVIEW

REVIEWER	Rothstein, Hannah The City University of New York, Baruch College Management
REVIEW RETURNED	21-Jul-2022

GENERAL COMMENTS	The authors report I squared, and tau squared, but they present I squared as their main index of heterogeneity. I squared is not an absolute measure of heterogeneity and should not be used as such. The authors must report the prediction interval in the results section and interpret those results properly in the discussion. Failure to do so will lead to misleading conclusions. Second, I am not sure why the authors used Cook's distance as a measure of outlying effect size, and only when it was significant did they use a leave one out analysis. Given what their forest plots look like, I believe they should proceed directly to a leave one out analysis. This would provide a better sensitivity test of the influence of single studies on the overall result of each analysis. Third, it would be useful to know whether the moderator(s) that were significant were a priori or post hoc. Given the number of moderators, and the K of studies, what is the likelihood that positive moderator results are due to capitalization on chance?
---

REVIEWER	Dadari, Ibrahim University of South Florida
REVIEW RETURNED	27-Jul-2022

GENERAL COMMENTS	Thank you for this important piece of work and for clarifying and responding to comments made as appropriate. This paper will be of great value!
--

REVIEWER	Di Ruggiero, Erica University of Toronto, Dalla Lana School of Public Health
REVIEW RETURNED	11-Jul-2022

GENERAL COMMENTS	The authors have addressed my feedback in a comprehensive. They may decide to add to the limitations section that it wasn't always clear how context was considered or measured in included studies (refer back to my comment about). I have no further comments.
---

REVIEWER	Wariri, Oghenebrume LSHTM
REVIEW RETURNED	18-Jul-2022

GENERAL COMMENTS	I would love to congratulate the authors for doing a great job in reviewing the manuscript based on the suggested edits. They have adequately addressed my comments, and I do not have further comments. I wish them well in their future research and endeavours.
--

VERSION 2 – AUTHOR RESPONSE

Reviewer: 1

Prof. Hannah Rothstein, The City University of New York

Comments to the Author:

The authors report I squared, and tau squared, but they present I squared as their main index of heterogeneity. I squared is not an absolute measure of heterogeneity and should not be used as such. The authors must report the prediction interval in the results section and interpret those results properly in the discussion. Failure to do so will lead to misleading conclusions.

Response - We have now included prediction intervals for our analyses, including those presented in appendices. We have also added this point to the limitation section of the discussion.

Second, I am not sure why the authors used Cook's distance as a measure of outlying effect size, and only when it was significant did they use a leave one out analysis. Given what their forest plots look like, I believe they should proceed directly to a leave one out analysis. This would provide a better sensitivity test of the influence of single studies on the overall result of each analysis.

Response - We used Cook's distances to identify overly influential studies with regard to the weight of the study, while we used studentized residuals to examine outliers (both as per Viechtbauer & Cheung, 2010). We added clarifying language around this point in the methods section. In addition, as requested, we conducted a leave-one-out analysis for each estimate and have added text reporting the results.

Viechtbauer, W. and Cheung, M.W.L., 2010. Outlier and influence diagnostics for meta-analysis. Research synthesis methods, 1(2), pp.112-125.

Third, it would be useful to know whether the moderator(s) that were significant were a priori or post hoc. Given the number of moderators, and the K of studies, what is the likelihood that positive moderator results are due to capitalization on chance?

Response - All moderators were decided a priori, with the exception of baseline vaccination coverage and vaccination hesitancy, which were added after feedback from initial peer review from the Campbell Collaboration (co-publisher of this work). This was noted in footnote ii on page 10.

Reviewer: 2

Dr. Ibrahim Dadari, University of South Florida

Comments to the Author:

Thank you for this important piece of work and for clarifying and responding to comments made as appropriate. This paper will be of great value!

Response - We thank the review for their thoughtful review of the initial work, and for the support of its' publication.

Reviewer: 3

Dr. Erica Di Ruggiero, University of Toronto

Comments to the Author:

The authors have addressed my feedback in a comprehensive. They may decide to add to the limitations section that it wasn't always clear how context was considered or measured in included

studies (refer back to my comment about). I have no further comments.

Response – We thank the reviewer for their thoughtful response. We would like to reiterate that we did not extract information related to the interesting questions posed by the reviewer on context, not necessarily that the questions could not be answered if we had done that. Therefore, we have decided not to add the point made by the reviewer to the limitation section.

Reviewer: 4

Dr. Oghenebrume Wariri, LSHTM

Comments to the Author:

I would love to congratulate the authors for doing a great job in reviewing the manuscript based on the suggested edits. They have adequately addressed my comments, and I do not have further comments. I wish them well in their future research and endeavours.

Response - We thank the review for their thoughtful review of the initial work, and for the support of its' publication.